# MultiDiffNet: A Multi-Objective Diffusion Framework for Generalizable Brain Decoding

## Abstract

Neural decoding from electroencephalography (EEG) remains fundamentally limited by poor generalization to unseen subjects, driven by high inter-subject variability and the lack of large-scale datasets to model it effectively. Existing methods often rely on synthetic subject generation or simplistic data augmentation, but these strategies fail to scale or generalize reliably. We introduce *MultiDiffNet*, a diffusion-based framework that bypasses generative augmentation entirely by learning a compact latent space optimized for multiple objectives. We decode directly from this space and achieve state-of-the-art generalization across various neural decoding tasks using subject and session disjoint evaluation. We also curate and release a unified benchmark suite spanning four EEG decoding tasks of increasing complexity (SSVEP, Motor Imagery, P300, and Imagined Speech) and an evaluation protocol that addresses inconsistent split practices in prior EEG research. Finally, we develop a statistical reporting framework tailored for low-trial EEG settings. Our work provides a reproducible and open-source foundation for subject-agnostic EEG decoding in real-world BCI systems.

## 1 Introduction

Electroencephalography (EEG) is a widely used modality in brain–computer interfaces (BCIs), supporting applications from assistive communication to cognitive monitoring. Deep learning has improved decoding across motor imagery, SSVEP, and speech tasks Gu et al. (2025); Ahmadi & Mesin (2025); Lee & Lee (2022), yet generalizing to unseen subjects remains challenging due to high inter-subject variability and limited data Huang et al. (2023); Barmpas et al. (2023).

Subject-specific models require extensive per-user calibration Hartmann et al. (2018); Luo & Cai (2024), while multi-subject models struggle to generalize Rommel et al. (2022); Liu et al. (2022); Wu (2016). The alternative is to use two-stage pipelines that generate EEG via GANs or diffusion and then train decoders (Hartmann et al., 2018; Torma & Szegletes, 2025), but they suffer from low realism, artifact transfer, and inefficiencies.

We propose *MultiDiffNet*, a unified multi-objective diffusion framework that learns a shared latent space, eliminating the need for synthetic augmentation and enhancing generalization. To benchmark progress, we release a curated suite spanning SSVEP, Motor Imagery, P300, and Imagined Speech tasks, with standardized subject- and session-disjoint evaluation. We also develop a statistical reporting protocol tailored for low-trial EEG research, addressing a persistent gap in reproducibility.

## 2 Related work

**EEG Decoding and Generalization** EEG decoding has evolved from handcrafted features to deep architectures, with EEGNet emerging as a widely adopted baseline due to its efficient depthwise–separable convolutions and lightweight design (Lawhern et al., 2018). Recent models explore transformers (Liao et al., 2025; Song et al., 2022a) and graph neural networks (Tang et al., 2024; Hu et al., 2023), but EEGNet remains favored for its robustness and simplicity. A key limitation is poor cross-subject generalization, with 20-40% accuracy drops despite strong within-subject perfor-

mance (Huang et al., 2023; Barmpas et al., 2023). Attempts to address this require expensive calibration (Rommel et al., 2022; Liu et al., 2022; Wu, 2016). Scalable BCIs require subject-agnostic models that generalize without per-user retraining.

**Diffusion Models for EEG** Denoising Diffusion Probabilistic Models (DDPMs) model data distributions via iterative denoising and outperform GANs in EEG synthesis by avoiding mode collapse (Tosato et al., 2023; Ho et al., 2020). Recent enhancements, such as reinforcement learning (An et al., 2024) and progressive distillation (Torma & Szegletes, 2025), have further improved realism and sampling speed. Diff-E (Kim et al., 2023) extended diffusion to imagined-speech decoding via joint reconstruction and classification, but remained task-specific and did not address cross-subject generalization. Broader research suggests that combining generative and discriminative objectives yields stronger representations (Chow et al., 2024; Grathwohl et al., 2019), yet EEG models typically optimize only one. We explore this joint learning paradigm across diverse EEG tasks, aiming to learn generalizable representations that capture both signal structure and task-relevant information.

**Mixup Methods** Signal-level augmentation has evolved from basic jittering and filtering to temporal, spectral, and channel-wise mixup (Luo & Cai, 2025; Liu et al., 2025; Kim et al., 2021; Pei et al., 2021; Zhang et al., 2017), but many variants introduce unrealistic artifacts that hinder generalization. This motivates our systematic evaluation of weighted and temporal input mixup across encoder layers, along with latent-space mixing

**Evaluation Strategies** Effective cross-subject EEG decoding requires both rigorous training strategies and standardized evaluation. Leave-one-subject-out (LOSO) validation remains common but is computationally intensive and impractical for real-time deployment (Del Pup et al., 2025; Chen et al., 2025; Zhao et al., 2024; Barmpas et al., 2023; Kunjan et al., 2021), while simpler subject splits often neglect session independence and true seen/unseen separation (Zhang et al., 2023). We address it in our work by introducing a standardized subject- and session-disjoint evaluation.

# 3 METHODOLOGY

## 3.1 MULTIDIFFNET ARCHITECTURE

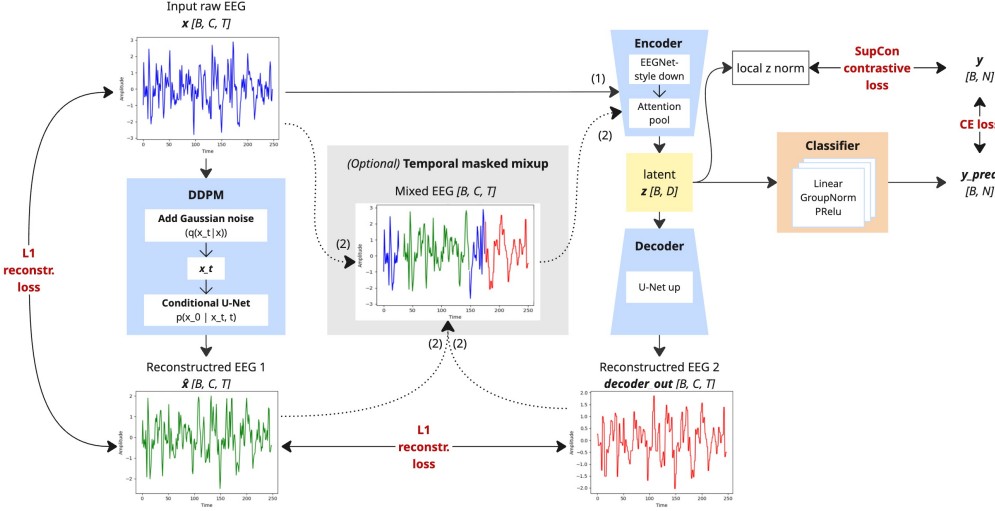

Figure 1: Overview of the *MultiDiffNet* that jointly optimizes a conditional DDPM, a contrastive encoder, and a generative decoder through a shared latent space *z*. The encoder produces discriminative features used for both classification and contrastive learning, while the decoder and DDPM reconstruct the input signal. An optional *temporal masked mixup* module stochastically blends the original, DDPM-denoised, and decoder-reconstructed EEG to improve representation quality.

*MultiDiffNet* is a modular architecture designed to jointly optimize classification, reconstruction, and contrastive structure learning from EEG signals. It consists of a Denoising Diffusion Probabilistic Model (DDPM), a discriminative encoder, a generative decoder, and a classifier (Figure 1).

Given a raw EEG signal $x \in \mathbb{R}^{C \times T}$, where $C$ is the number of EEG channels and $T$ is the number of timepoints, the model processes the input in two parallel paths. First, the DDPM denoises the signal via a learned reverse diffusion process, producing a refined version $\hat{x} \in \mathbb{R}^{C \times T}$. Simultaneously, the same input $x$ is passed through an EEGNet-based encoder (See Section 3.2) to extract a latent representation $z \in \mathbb{R}^D$, where $D$ is the embedding dimension. The latent vector $z$ is then used for two purposes: (1) it is passed to a lightweight decoder to reconstruct the denoised signal $\hat{x}$, resulting in a reconstruction $x_{\text{dec}} \in \mathbb{R}^{C \times T}$; and (2) it is passed to a fully connected classification head to predict class logits $\hat{y} \in \mathbb{R}^K$, where $K$ is the number of classes.

To further structure the latent space, $z$ is locally normalized (Section 3.3) and then projected to $z_{\text{proj}} \in \mathbb{R}^{D'}$, which is optimized with a supervised contrastive loss. All classification and reconstruction are performed directly from $z$, without relying on generated augmentations.

We performed an extensive ablation study across architectural variants, modifying the presence of DDPMs, encoder inputs, decoder pathways, classifier heads, and loss terms. The configuration described here reflects the best-performing combination.

## 3.2 EEGNet-style encoder with attention pool

Given EEGNet's demonstrated effectiveness across multiple EEG decoding tasks, we adapt its architecture as our discriminative encoder, hypothesizing that its proven feature extraction capabilities can produce powerful latent representations $z$ for our multi-objective framework. Our encoder extracts multi-scale features $(dn_1, dn_2, dn_3)$ from different layers and applies attention pooling:

$$z = \text{AttentionPool}(dn_3) \in \mathbb{R}^D,$$

## 3.3 Subject-wise latent normalization

To mitigate inter-subject variability, we apply subject-wise normalization on the encoder output $z$:

$$z_{\text{norm}} = \frac{z - \mu_s}{\sigma_s},$$

where $\mu_s$ and $\sigma_s$ denote the mean and standard deviation computed per subject $s$ using a subset of training trials. During evaluation, we adopt a two-mode strategy: for seen subjects, normalization uses pre-computed statistics from their training data; for unseen subjects, statistics are estimated on-the-fly using their own calibration trials, simulating realistic deployment scenarios.

## 3.4 Mixup strategies

Mixup strategies can improve robustness in low-trial EEG decoding. However, standard mixup techniques may not fully exploit the structure of neural time series. We therefore explore two complementary strategies: *Weighted Average Mixup* and a novel *Temporal Masked Mixup*. *Weighted Average Mixup* performs linear interpolation between the original EEG input $x$, the DDPM-denoised output $\hat{x}$, and the decoder reconstruction $x_{\text{dec}}$. We investigate multiple integration points in the model: **(0)** Input-level mixup, **(1-3)** Mixup after encoder layers 1, 2, or 3, respectively, **(4)** Mixup after the final attention pooling layer. To address the limitations of global interpolation, we propose *Temporal Masked Mixup*, which perturbs only localized segments of the input time series while preserving surrounding structure. See Algorithm 1 for pseudocode.

## 3.5 Loss functions

*MultiDiffNet* is trained using a weighted sum of three objectives:

$$\mathcal{L}_{\text{total}} = \underbrace{\alpha\, \mathcal{L}_{\text{CE/MSE}}(\hat{y}, y)}_{\text{classification}} + \underbrace{\beta\, \mathcal{L}_{\text{L1}}(x_{\text{dec}}, \hat{x})}_{\text{reconstruction}} + \underbrace{\gamma\, \mathcal{L}_{\text{SupCon}}(z_{\text{proj}}, y)}_{\text{contrastive}}$$

---

**Algorithm 1** Temporal Masked Mixup

---

1: Initialize a binary mask $M \in \{0,1\}^{C \times T}$ with all zeros.
2: Flip each 0 in $M$ to 1 with probability $p = 0.01$.
3: **for** each position in $M$ with value 1 **do**
4:     Expand to a temporal window of random length (uniform between min and max size).
5: **end for**
6: Flip each 1 in $M$ to $-1$ with:
- Fixed probability 0.5 (**fixed ratio**), or
- Probability drawn from Beta$(0.2, 0.2)$ each epoch (**random ratio**).
7: Apply the final mask:
- $0 \rightarrow x$ (original input)
- $1 \rightarrow \hat{x}$ (DDPM output)
- $-1 \rightarrow x_{\text{dec}}$ (decoder output)

---

We fix $\alpha = 1.0$ and progressively scale $\beta$ and $\gamma$ to stabilize training:

$$\beta = \min\left(1.0, \frac{\text{epoch}}{100}\right) \cdot 0.05, \quad \gamma = \min\left(1.0, \frac{\text{epoch}}{50}\right) \cdot 0.2$$

Details on loss formulation and weighting strategies are provided in the Appendix.

### 3.6 EVALUATION METRICS

We evaluate model performance primarily using downstream classification accuracy, which quantifies the proportion of correctly classified EEG samples. Accuracy is defined as:

$$\text{Accuracy} = \frac{TP + TN}{TP + TN + FP + FN}$$

where $TP$, $TN$, $FP$, and $FN$ denote true positives, true negatives, false positives, and false negatives, respectively. In addition, we report F1 score, precision, recall, and AUC for a more comprehensive evaluation; detailed formulas and results are provided in the Appendix.

### 3.7 TREND-LEVEL STATISTICAL REPORTING FRAMEWORK

Conventional $p$-values often fail under the high-variance, low-trial, subject-disjoint conditions of EEG decoding. To address this, we introduce a robust trend-level statistical framework (detailed in the Appendix) that synthesizes effect sizes, cross-seed consistency, and Bayesian posterior probabilities. This allows us to detect systematic, reproducible gains even when classical significance tests return null results. Our approach represents a principled shift toward reproducible, evidence-based model evaluation in brain decoding.

While this framework enhances reproducibility, it is not meant to substitute conventional $p$-value testing. Instead, it addresses a well-documented limitation: in low-trial, high-variance EEG decoding, even systematic improvements may fail to reach arbitrary significance thresholds. Such small yet consistent gains—for instance, 2–3% accuracy in imagined speech or SSVEP—can substantially affect usability in BCI systems. By combining effect sizes, cross-seed consistency, and Bayesian evidence, the framework provides a principled way to surface these domain-relevant improvements, while remaining fully compatible with classical and non-parametric statistical tests.

## 4 EXPERIMENTS AND RESULTS

### 4.1 BENCHMARK DATASET SUITE

We curated four diverse EEG benchmarks (SSVEP, P300, Motor Imagery, and Imagined Speech), spanning increasing decoding difficulty. Each dataset is split into train, val, and two test sets: a seen-subject (intra-subject) split and an unseen-subject (cross-subject) split. This standardized protocol

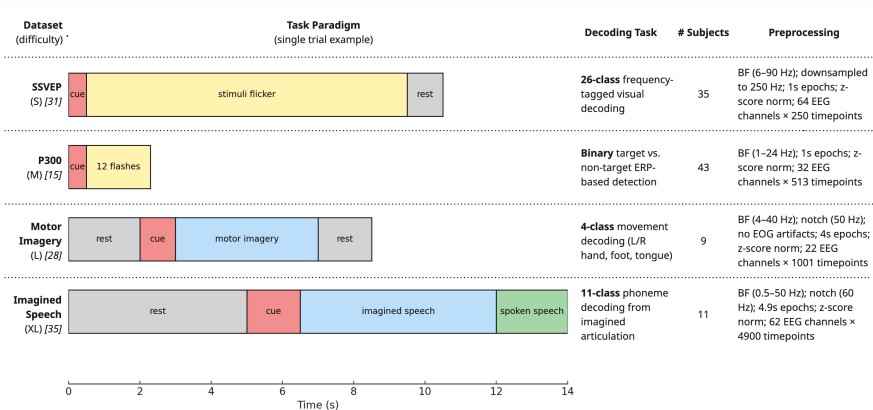

Figure 2: Overview of four EEG datasets ranked by task difficulty from easiest (top) to hardest (bottom). Task paradigms and preprocessing details are adapted from the original publications: SSVEP Wang et al. (2017), P300 Korczowski et al. (2019), Motor Imagery Tangermann et al. (2012), and Imagined Speech Zhao & Rudzicz (2015).

enables rigorous evaluation of both personalization and generalization, addressing the inconsistent and often unrealistic split practices prevalent in prior EEG research, where models are evaluated on mixed subject data or using computationally expensive LOSO.

## 4.2 BASELINES

We benchmarked our model against a diverse set of carefully selected baselines to ensure robust and fair comparisons. Our selection criteria were twofold: (i) prioritize architectures that are widely used for generalization to unseen subjects or sessions, and (ii) cover the main inductive biases found in EEG decoding, such as spatial filtering, temporal modeling, and attention mechanisms.

Specifically, we include: (1) **EEGNet** (Lawhern et al., 2018), a compact depthwise-separable CNN that is widely adopted for cross-subject generalization due to its strong accuracy–efficiency trade-off; (2) **ShallowFBCSPNet** (Schirrmeister et al., 2017), which implements learnable filter-bank Common Spatial Patterns (CSP) to extract frequency–spatial features; (3) **TIDNet** (Kostas & Rudzicz, 2020), which introduces dilated convolutions and residual connections to improve robustness under subject shift; (4) **EEGConformer** (Song et al., 2022b), which combines a convolutional front-end with self-attention to model both local spatial structure and global temporal context; and (5) **EEGTCNet** (Ingolfsson et al., 2020), a temporal convolutional network tailored for EEG that emphasizes causal and dilated temporal modeling, offering complementary inductive bias to purely spatial–spectral models.

All models are evaluated using identical input windows of shape $(C, T)$, and trained with a unified global training schedule to ensure comparability. Public implementations and recommended hyperparameters are used where available, with no method-specific tuning.

## 4.3 GENERALIZATION PERFORMANCE

*MultiDiffNet* helps with generalization. Unlike raw EEG representations, where class boundaries blur due to subject-specific noise, our learned latent space forms clearly separable, label-aligned clusters (Figure 3). This structured representation enables robust decoding across subjects. As shown in Table 1, *MultiDiffNet* consistently reduces the seen–unseen accuracy gap across all tasks. In SSVEP, it lifts cross-subject accuracy from 81.08% (EEGNet baseline) to 84.72%, further boosted to 85.25% with Temporal Masked Mixup. For comparison, other representative architectures such as ShallowFBCSPNet (58.87%), EEGConformer (51.92%), TIDNet (25.96%), and EEGTCNet (49.57%) fall well behind, highlighting the robustness of our latent-space design.

Even in the low-SNR regime of Imagined Speech, *MultiDiffNet* improves cross-subject accuracy from 10.61% (EEGNet) to 12.12%, while simultaneously achieving a much larger gain on seen-

Table 1: Final results across tasks and models. Accuracy is reported for both seen-subject (intra-subject) and unseen-subject (cross-subject) test splits. Tasks are ranked by task difficulty. *Stars denote win percentage:* ***\*\*\* $\geq$ 80%, \*\* $\geq$ 60%, \* $\geq$ 40%.*** Detailed results are in the Appendix.

| Task | Model | Subj. | Classes | Seen Acc. (%) | Unseen Acc. (%) |
|------|-------|-------|---------|---------------|-----------------|
| **SSVEP** | ShallowFBCSPNet | 35 | 26 | $69.58 \pm 1.30^{*}$ | $58.87 \pm 9.37^{*}$ |
| | EEGConformer | 35 | 26 | $66.98 \pm 2.83$ | $51.92 \pm 9.06$ |
| | TIDNet | 35 | 26 | $28.01 \pm 4.12$ | $25.96 \pm 5.29$ |
| | EEGTCNet | 35 | 26 | $58.31 \pm 4.02$ | $49.57 \pm 9.14$ |
| | EEGNet | 35 | 26 | $89.16 \pm 0.57^{***}$ | $81.08 \pm 9.16^{**}$ |
| | **MultiDiffNet** | 35 | 26 | $85.08 \pm 1.53^{**}$ | $\mathbf{84.72 \pm 6.03}^{***}$ |
| | **MultiDiffNet + Mixup** | 35 | 26 | $86.79 \pm 1.75^{***}$ | $\mathbf{85.25 \pm 6.94}^{***}$ |
| **P300** | ShallowFBCSPNet | 43 | 2 | $87.72 \pm 0.33$ | $86.20 \pm 1.45$ |
| | EEGConformer | 43 | 2 | $88.54 \pm 0.54^{**}$ | $86.30 \pm 1.73$ |
| | TIDNet | 43 | 2 | $88.24 \pm 0.31^{*}$ | $85.63 \pm 0.58^{**}$ |
| | EEGTCNet | 43 | 2 | $88.69 \pm 0.59^{***}$ | $87.02 \pm 1.62^{***}$ |
| | EEGNet | 43 | 2 | $88.79 \pm 0.67^{***}$ | $87.24 \pm 2.01^{***}$ |
| | **MultiDiffNet** | 43 | 2 | $85.35 \pm 1.12$ | $79.47 \pm 0.54^{*}$ |
| | **MultiDiffNet + Mixup** | 43 | 2 | $85.61 \pm 0.52$ | $79.56 \pm 4.43$ |
| **MI** | ShallowFBCSPNet | 9 | 4 | $64.34 \pm 3.61^{***}$ | $36.46 \pm 6.60$ |
| | EEGConformer | 9 | 4 | $59.57 \pm 5.60^{**}$ | $36.49 \pm 7.72$ |
| | TIDNet | 9 | 4 | $44.27 \pm 2.60$ | $34.42 \pm 3.60$ |
| | EEGTCNet | 9 | 4 | $58.85 \pm 4.54$ | $32.99 \pm 6.94$ |
| | EEGNet | 9 | 4 | $67.01 \pm 5.38^{***}$ | $46.18 \pm 7.20^{***}$ |
| | **MultiDiffNet** | 9 | 4 | $55.85 \pm 2.80$ | $39.24 \pm 8.00^{***}$ |
| | **MultiDiffNet + Mixup** | 9 | 4 | $57.69 \pm 3.27^{*}$ | $36.78 \pm 5.23$ |
| **Img. Speech** | ShallowFBCSPNet | 11 | 11 | $13.78 \pm 1.55^{**}$ | $10.48 \pm 0.64$ |
| | EEGConformer | 11 | 11 | $10.62 \pm 0.82$ | $9.21 \pm 3.00$ |
| | TIDNet | 11 | 11 | $9.10 \pm 0.54$ | $10.35 \pm 0.18$ |
| | EEGTCNet | 11 | 11 | $12.64 \pm 1.58$ | $10.10 \pm 0.64$ |
| | EEGNet | 11 | 11 | $11.26 \pm 2.01^{*}$ | $10.61 \pm 0.93^{*}$ |
| | **MultiDiffNet** | 11 | 11 | $\mathbf{15.55 \pm 0.62}^{***}$ | $\mathbf{11.62 \pm 1.29}^{***}$ |
| | **MultiDiffNet + Mixup** | 11 | 11 | $\mathbf{17.57 \pm 1.16}^{***}$ | $\mathbf{12.12 \pm 0.38}^{***}$ |

subject accuracy (11.26% → 17.57%). Other baselines such as ShallowFBCSPNet (10.48/13.78%), EEGConformer (9.21/10.62%), TIDNet (10.35/9.10%), and EEGTCNet (10.10/12.64%) hover close to chance level on both splits, further highlighting the robustness of our approach. For such a challenging task, even modest absolute gains are meaningful, as they can indicate more reliable signal extraction under extreme noise conditions. On Motor Imagery, *MultiDiffNet* also surpasses most baselines on unseen accuracy, e.g., outperforming TIDNet (34.42%) and EEGTCNet (32.99%), while maintaining competitive seen accuracy (57.69% vs. 44.27% for TIDNet and 58.85% for EEGTCNet). Although it remains slightly below EEGNet (46.18/67.01%), this is likely due to ceiling effects and dataset scale.

## 4.4 ABLATION STUDIES

To understand what drives generalization in *MultiDiffNet*, we ran extensive ablation experiments, over 100 controlled configs. All results are reported for both seen- and unseen-subject accuracy, with statistical evidence matrices and trend-level effect sizes in the Appendix.

**Decoder input.** Feeding only $z$ to the decoder often matches or exceeds more complex fusion variants. For example, SSVEP unseen accuracy reaches 84.72% with $z$ alone, further boosted to 85.25% with mixup, while more elaborate fusions ($z + x$, $x_{hat}+$ skips) show no consistent gains. These findings validate our architectural decision to decode primarily from $z$. For completeness, the best

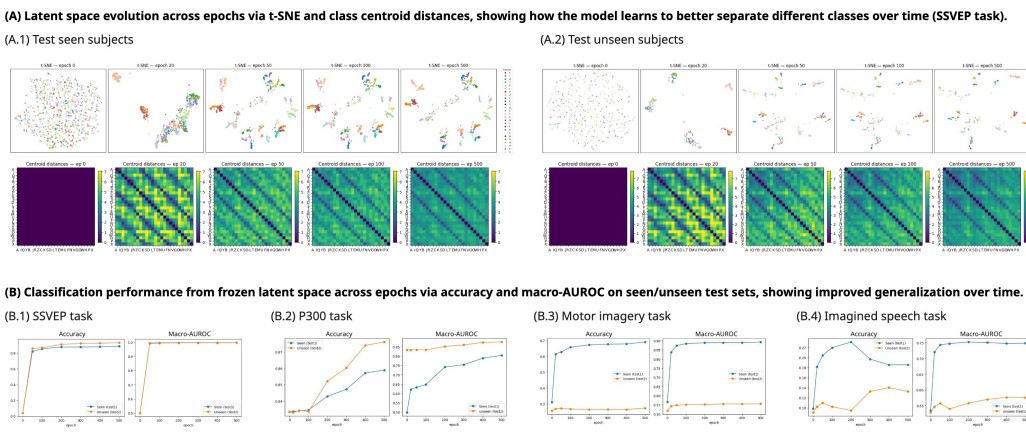

Figure 3: (A) Visualization of latent space across training epochs. (B) Downstream classification performance from frozen latent representations.

accuracies achieved in this ablation are 85.86/84.72 on SSVEP, 85.88/81.41 on P300, 56.89/40.36 on Motor Imagery, and 18.58/12.88 on Imagined Speech (seen/unseen).

**Classifier head.** A lightweight FC head on $z$ delivers state-of-the-art generalization with minimal complexity. It rivals or outperforms EEGNet classifiers trained on $x$, especially in low-SNR tasks. This supports our choice to use FC as the default classification head. For completeness, the best accuracies achieved in this ablation are 85.08/84.72 on SSVEP, 85.35/84.12 on P300, 55.85/39.24 on Motor Imagery, and 17.95/11.61 on Imagined Speech (seen/unseen).

**Encoder and decoder.** Using raw $x$ as encoder input consistently outperforms $\hat{x}$, showing that denoising is useful for regularization. Interestingly, removing the decoder entirely sometimes improves generalization, suggesting that reconstruction may introduce noise if overemphasized. For completeness, the best accuracies in this ablation are 90.95/85.58 on SSVEP, 85.71/80.93 on P300, 55.85/40.16 on Motor Imagery, and 19.22/13.76 on Imagined Speech (seen/unseen).

**Loss combinations.** Combining CE with mild MSE or contrastive losses improves stability, particularly when auxiliary weights are gently annealed. The best results use $\beta = 0.05, \gamma = 0.2$—balancing reconstruction as a regularizer without overpowering the classification objective. For completeness, the best accuracies in this ablation are 86.40/85.58 on SSVEP, 85.69/80.18 on P300, 59.67/41.44 on Motor Imagery, and 19.60/13.51 on Imagined Speech (seen/unseen).

**Mixup strategies.** Mixup effects are task-specific. For SSVEP, *Temporal Masked Mixup* outperforms all variants. Motor Imagery benefits from *Weighted Average Mixup*, while P300 and Imagined Speech show limited sensitivity, highlighting that mixup is most impactful in high-SNR regimes. For completeness, the best accuracies in this ablation are 87.84/85.26 on SSVEP, 85.78/79.56 on P300, 63.44/38.83 on Motor Imagery, and 19.47/12.12 on Imagined Speech (seen/unseen).

## 5    CONCLUSIONS AND FUTURE WORK

We presented *MultiDiffNet*, a diffusion-based neural decoder that learns a compact, multi-objective latent space for EEG decoding without synthetic augmentation. Through unified benchmarks and rigorous cross-subject evaluation, we showed that *MultiDiffNet* achieves strong generalization across diverse BCI paradigms, particularly in challenging low-signal settings such as SSVEP and Imagined Speech. Our statistical analysis framework further addresses reproducibility challenges in low-trial EEG research. Future work will explore scaling *MultiDiffNet* to larger and more diverse EEG datasets and extending the architecture to other neural modalities.

For completeness, we note that our trend-level statistical framework is intended only as a complementary tool for low-trial EEG research; detailed rationale is provided in Section 3.7, with Bayesian and non-parametric validations reported in the Appendix.

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
