# OpenReview forum: "MultiDiffNet: A Multi-Objective Diffusion Framework for Generalizable Brain Decoding"
_ICLR.cc/2026/Conference — Submitted to ICLR 2026_

### Official Review · Reviewer_omEe · 2025-10-15

**Soundness:** 2
**Presentation:** 3
**Contribution:** 2
**Rating:** 2
**Confidence:** 4

**Summary:**

This paper introduces MultiDiffNet, a complex, diffusion-based framework for EEG decoding. The authors' goal is to improve generalization across unseen subjects by learning a latent space optimized for classification, reconstruction, and contrastive objectives, thereby avoiding the need for explicit data augmentation. The paper also presents a new benchmark suite of four EEG tasks with standardized data splits and proposes a new "trend-level" statistical framework for evaluating results in low-trial settings. The central claim is that this approach achieves state-of-the-art generalization.

**Strengths:**

The paper's most significant and durable contribution is the curation and release of a standardized benchmark suite for EEG decoding . Establishing a rigorous, subject-disjoint evaluation protocol is a valuable service to the community that will foster more reproducible research.
The authors have conducted an exceptionally detailed ablation study, investigating over 100 configurations.

**Weaknesses:**

The central narrative of achieving superior generalization is not supported by the paper's own results. In 50% of the tasks (P300 and Motor Imagery), simpler baselines outperform MultiDiffNet on the crucial unseen-subject test set. The proposed complex model does not justify its existence with consistent, clear-cut performance gains.
The proposed MultiDiffNet architecture is substantially more complex than the baselines it fails to consistently outperform. The ablation study reveals that removing the decoder, a core component of the multi-objective framework, can actually improve generalization, suggesting the authors' own understanding of the model's mechanics is incomplete. The added complexity does not bring a corresponding, reliable benefit.

**Questions:**

1) Given that your model is outperformed by simpler baselines on half of the benchmark tasks, how can you maintain the claim of achieving "state-of-the-art generalization"? Please provide a more nuanced and accurate assessment of your model's capabilities and limitations.

2) The finding that removing the decoder can improve performance severely undermines the paper's premise that a reconstruction objective is key to learning a generalizable latent space. How do you explain this result, and what does it imply about the actual source of any performance gains you do observe?

3) Could you provide a compelling argument for why the community should adopt a significantly more complex model that offers, at best, task-specific performance improvements over simpler, faster, and more established architectures like EEGNet?

---

### Official Review · Reviewer_n1ee · 2025-10-31

**Soundness:** 1
**Presentation:** 1
**Contribution:** 2
**Rating:** 2
**Confidence:** 4

**Summary:**

The paper introduces a new model for decoding brain signals.  The method proposes a three-loss-based model using classification loss, reconstruction loss, and contrastive loss. Then they compare their method with five different models over 4 EEG tasks.

**Strengths:**

The paper proposes tackling generalization for the EEG task, which is a challenging yet important task.

**Weaknesses:**

The paper lacks clarity:
- Figure 3 is unreadable; all the captions and axis labels are too small, making it hard to understand what we are looking at.
- The appendix is hard to follow, but all the ablation study is inside it. Since there are two additional pages available, it would be easier to follow if the ablation study were included within the main paper.
- In the results table, bold is used only when the method's author is the best, and no bold is used when the proposed method is worse than competitors. making the table hard to read. Additionally, MultiDiffNet suffers a considerable performance loss on P300 and MI. 8% and 7% respectively. If the text says "slightly" below, I think it represents a significant decrease.
- The introduction is very short. Since there is some space available, it could be interesting to have a better introduction to the literature to provide stronger motivation. Right now, there is no clear motivation for the paper.

If the paper's pipeline is interesting, improving the writing and clarity will enhance the contribution.

**Questions:**

Why the score for Imagined speech is so low?

---

### Official Review · Reviewer_hFQT · 2025-10-31

**Soundness:** 3
**Presentation:** 2
**Contribution:** 2
**Rating:** 4
**Confidence:** 3

**Summary:**

MultiDiffNet improves cross-subject generalization on several EEG decoding tasks using a latent space trained jointly for classification, reconstruction, and contrastive objectives. On SSVEP, unseen-subject accuracy rises from 81.08% with EEGNet to 84.72%, and to 85.25% with Temporal Masked Mixup. On imagined speech, it lifts seen-subject accuracy from 11.26% to 17.57% and unseen from 10.61% to 12.12%. Motor imagery shows competitive seen accuracy and better unseen accuracy than several baselines, though still below EEGNet on seen splits. P300 results lag strong baselines. The model consistently narrows the seen-vs-unseen gap, and extensive ablations identify which components matter most.

Contributions. The paper introduces a multi-objective diffusion framework that avoids synthetic augmentation by learning a compact shared latent space with an EEGNet-style encoder, subject-wise latent normalization, a lightweight classifier head, and two mixup strategies including a new Temporal Masked Mixup. It also releases a unified benchmark across four tasks with standardized subject- and session-disjoint splits, and proposes a trend-level statistical reporting scheme tailored to low-trial EEG to surface small but consistent gains. Together these pieces form a reproducible setup for subject-agnostic EEG decoding and clarify evaluation practices.

**Strengths:**

Originality: Joint multi-objective diffusion with subject-wise latent normalization and Temporal Masked Mixup for cross-subject EEG.
Quality: Standardized subject/session-disjoint protocol, strong baselines under one schedule, plus thorough ablations and trend-level stats.
Clarity: Architecture and objectives are explicit; losses, mixup algorithm, and normalization are clearly defined.
Significance: Delivers unseen-subject gains on SSVEP and imagined speech; provides a reusable benchmark and reporting scheme.

**Weaknesses:**

1 Claims do not match results. The paper says it consistently narrows the seen vs unseen gap across tasks, yet on P300 and Motor Imagery it trails EEGNet and the gap can widen.

2 Subject-wise normalization may bias comparisons. You normalize the latent space with per-subject statistics. For unseen users it is unclear how those statistics are obtained. If any calibration trials are used, every baseline must get the same treatment.

3 The paper labels the classification loss “CE/MSE” without stating which is used in the main results. The reconstruction aligns the decoder to the DDPM output rather than the raw signal; if that target is updated jointly and not frozen, training can chase a moving target. State the exact loss used, whether the denoised target is gradient-detached, and the training order.

4. Accuracy is presented with a binary confusion-matrix formula while some tasks are multi-class, and the “win rate” stars are not defined. Provide a clear multi-class metric definition, add macro and micro F1, define the unit of a “win” and the pairing, and include confidence intervals plus a formal test or Bayes factor for the key head-to-head claims.

**Questions:**

1. when authors say the method consistently narrows the seen vs unseen gap across tasks. The tables show mixed outcomes, especially on P300 and Motor Imagery. Please explain, and it would be better to add a short failure-mode analysis for those two tasks with concrete fixes you tried.

2.For unseen subjects, how did authors obtain their per-subject statistics? If any calibration trials are used, every baseline should get the same procedure. Please report two settings for all methods: zero-calibration and few-shot calibration.

3 What exactly is the loss and training flow?
State which classification loss you used in the main results. Clarify what the reconstruction is aligned to, and whether the denoised target was kept fixed during decoder updates. A small stability plot comparing “frozen target” vs “jointly updated target” would settle concerns.

4 Are the evaluation definitions rigorous and transparent?
Several tasks are multi-class, yet accuracy is described with a binary definition and “win rate” is not clearly defined. Please add the exact multi-class metric, report macro and micro F1 alongside accuracy, define what counts as a “win,” and include confidence intervals plus a simple significance or Bayesian test for the key head-to-head claims.

**Details Of Ethics Concerns:**

The current ethics statement is insufficient as it fails to address core concerns like the validity of informed consent for aggregated data reuse and the potential for misuse of the neural decoding technology.

---

### Official Review · Reviewer_KQi5 · 2025-11-03

**Soundness:** 2
**Presentation:** 2
**Contribution:** 1
**Rating:** 0
**Confidence:** 4

**Summary:**

This paper relates to generalizable representation learning for brain EEG date. The objective is to learn well separable clusters, even for unseen data. They go about solving this problem by having a diffusion module, an encoder, a decoder, and a classifier. Additionally, they have 4 different losses:
- 1.Recon losses:
  - a. A simple diffusion based gaussian denoising/recon. Loss
  - b. Labeled and unlabeled mixing and recon loss
- 2. Encoder outputs
  - a. Contrastive Supcon Loss
  - b. Classification loss

The results show that their work performs really well on unseen data, especially by having separable clusters.

**Strengths:**

They show decent results on their chosen datasets and beat a lot of SOTA results by a decent margin, especially on the unseen categories.

**Weaknesses:**

- The paper is very procedural: It is about applying some already very well known techniques via multiple losses and techniques. There is no hypothesis that they are trying to prove or disapprove; It’s a benchmarking paper at best. On the theoretical side they have combined a lot of well-known losses. But no questions have been answered like why this way is the best way to solve the given problem. They claim that other generative synthetic augmentations are not scalable, but no computational-time-cost vs performance benchmarks were provided.

- This paper has a wording problem: What do they exactly mean by compactness? Is it related to compression, rate-distortion stuff? Or do they mean that the classes are well separable and class prototypes are compact? For the latter, they haven’t compared the separability of their work against other and shown that their is better at separation. Showing better performance doesn’t mean that the class-separation is better as well.

- The paper is only 7 pages, which although not a requirement, shows that a lot more thoughtful experiments could be added.

**Questions:**

None.

---

### Official Review · Reviewer_LGZ7 · 2025-11-05

**Soundness:** 1
**Presentation:** 2
**Contribution:** 2
**Rating:** 2
**Confidence:** 4

**Summary:**

The work presents a model named MultiDiffNet, an EEG decoder that aims to learn a compact and structured latent space for EEG decoding with multi-objective training. The idea is to use the EEG signal generated from a diffuser and the reconstructed signal from the latent representation via a decoder to mix with the original EEG signal in mixup fashion. Such augmentation will avoid other synthetic augmentations that have the tendency to result in unreal EEG. The model is trained to jointly optimize both EEG generation/reconstruction (diffusion, and decoder), downstream classification, and supervised contrastive loss. Evaluation was carried out on 4 different EEG decoding tasks.

**Strengths:**

The idea of using reconstructed EEGs (from diffusion, decoder-based reconstruction) for augmentation is fine, particularly for EEGs, as one could expect the reconstructed EEGs are less subject to subject-specific noise and eventually improve generalization across subjects.

However, this idea is not new, but the proposal to jointly train the diffusor & the decoder for EEG generation/reconstruction jointly with the encoder in multi-objective is.

**Weaknesses:**

{\bf Method:}

The approach that trains jointly generation/reconstructor and the encoder seems overkilled. I understand the motivation to unify them in multi-objective training, but I do not see the benefits in terms of performance and generalization (there are no experiments to showcase that). At the same time, I am also concerned about the stability of training such a model. In the early phase, when the diffusion and reconstructor are not yet in good state, the reconstructed EEGs are bad, would they be any good for data augmentation? The design of loss scheduler may mitigate this with different weights on different components, but I don't think it would resolve this issue.

The so-called temporal masked mixup that is claimed to be new in the work is actually similar to CutMix augmentation, which I don't think is considered novel.

{\bf Evaluation:}

In my opinion, the experimental design for evaluation is not yet appropriate to showcase the advantage of the work. As the work in fact focuses on generating EEGs for augmentation, I would expect comparison with existing augmentation methods to showcase its advantage. The unified multi-objective training also warrants the comparison with two-stage training.

I am not convinced with the choices for baselines. Most of them are >5 years old and do not represent the recent advance in this topic. Also, I wonder if training these baselines without "method-specific tuning" is a reasonable way for a fair comparison. Different models and architectures may favor different training procedures and one-for-all receipt will hide the true potential of a model.

I appreciate the effort in standardizing evaluation which I believe is important for reproducible research. However, I would defense for LOSO evaluation which is probably better suit for academic setting & small datasets like these. Particularly, the amount of training data will be maximized with LOSO, which is important in deep learning era. It also facilitates the reproducibility and comparability between different works as well.

I don't know how to interpret the claim "state-of-the-art generalization". Is the the gap between seen-subject and unseen-subject performance or the absolute performance themselves? In any case, neither of the interpretations is backed by the results in Table 1. The results show a mix where the proposed method wins on SSVEP and Imagined Speech, but it underperforms other methods on MI and P300 tasks, particularly the latter showing a big gap on unseen-subject classification. I am afraid that one could draw a conclusion confidently from these results.

**Questions:**

- What are the benefits of multi-objective training over two-stage training, particularly in terms of performance and generalization?
- Could you comment on the stability of the training? In the early phase, when the diffusion and reconstructor are not yet in good state, the reconstructed EEGs are bad, would they be any good for data augmentation?
- How the loss weight scheduler was design given that the weight for diffusion and contrastive training is much smaller than the classification loss?
- Given most of the baselines are >5 years old, how one could position the results in this work in literature, particularly the SOTA on these decoding tasks?
- Would the way the baselines models were trained ensure fair comparison? And how?
- How could we interpret the mix results in Table 1 to support the claim "SOTA generalization"?

---

### Official Review · Reviewer_gmfV · 2025-11-08

**Soundness:** 2
**Presentation:** 2
**Contribution:** 2
**Rating:** 4
**Confidence:** 3

**Summary:**

This paper presents an approach to addressing a key challenge in EEG-based brain-computer interfaces: poor generalization across subjects due to inter-subject variability and limited data. The proposed MultiDiffNet framework integrates diffusion models (DDPM) with a shared latent space optimized for classification, reconstruction, and contrastive learning. It also provided a curated benchmark suite across four EEG tasks (SSVEP, P300, Motor Imagery, and Imagined Speech) with standardized subject/session-disjoint splits.

**Strengths:**

The proposed approach avoids common pitfalls like artifact introduction in GAN-based or diffusion-based synthesis. The Temporal Masked Mixup is a nice extension of standard mixup, preserving temporal structure in EEG signals.

By standardizing datasets and enforcing subject/session-disjoint splits, it addresses inconsistencies in prior works.

The trend-level statistical framework mitigates p-value limitations in high-variance settings, promoting evidence-based claims.

Ablations covering decoder inputs, classifier heads, encoder/decoder variants, loss combinations, which is relatively comprehensive.

**Weaknesses:**

No runtime or parameter count comparisons; given EEGNet's lightweight design, how does MultiDiffNet's added DDPM/decoder components affect efficiency for real-time BCIs?

Some hyperparameters such as embedding dimensions, attention pool specifics are underspecified in the main text. This could hinder replication. Mixup integration points are mentioned but results aren't fully tabulated, which point works best per task?

P300 results are mixed with MultiDiffNet underperforms baselines on unseen accuracy. The paper attributes this to "ceiling effects," more discussion is needed.

**Questions:**

Please see above.

---

### Meta-Review · Area_Chair_4sCk · 2026-01-05

**Summary:**

The paper proposes a framework using diffusion models to overcome inter-subject variability and limited data. Evaluation is performed on four tasks.

Strength: The work tackles a challenging issue in EEG. It tries to standardize evaluation and datasets.

Weakness:

(1) Most reviewers indicate that the claim of improved generalization performance is not supported well. The performance of the proposed method is behind other methods in a half of the tasks.

(2) The novelty is questioned by two reviewers.

(3) A reviewer raises an issue about how per-subject normalization can be done also for unseen subjects and whether any calibration trials were applied to all methods.

(4) Several reviewers mention clarity issues.

**Reviewer Concerns:**

No rebuttal was submitted.

**Reviewer Scores:**

No rebuttal was submitted. All the six reviewers provide negative ratings. No conflicting reviewer comments are found. The reviewers wouldn't have changed their scores through discussion.

---

### Decision · Program_Chairs · 2026-01-26

Reject